# A Simplified Measurement Configuration for Evaluation of Relative Permittivity Using a Microstrip Ring Resonator with a Variational Method-Based Algorithm

**DOI:** 10.3390/s22030928

**Published:** 2022-01-25

**Authors:** Miroslav Joler, Alex Noel Joseph Raj, Juraj Bartolić

**Affiliations:** 1Department of Computer Engineering, Faculty of Engineering, University of Rijeka, 51000 Rijeka, Croatia; mjoler@riteh.hr; 2Department of Electronic Engineering, College of Engineering, Shantou University, Shantou 515063, China; 3Department of Wireless Communications, Faculty of Electrical Engineering and Computing, University of Zagreb, 10000 Zagreb, Croatia; juraj.bartolic@fer.hr

**Keywords:** measurement of relative permittivity, full-wave analysis, microstrip ring resonator, optimal material-under-test size, material characterization, variational method

## Abstract

In this paper, we present a simple yet efficient method for determination of the relative permittivity of thin dielectric materials. An analysis that led to definition of the proper size and placement of a sample under test (SUT) on the surface of a microstrip ring resonator (MRR) was presented based on the full-wave simulations and measurements on benchmark materials. For completeness, the paper includes short descriptions of the design of an MRR and the variational method-based algorithm that processes the measured values. The efficiency of the proposed method is demonstrated on 12 SUT materials of different thicknesses and permittivity values, and the accuracy between 0% and 10% of the relative error was achieved for all SUTs thinner than 2 mm.

## 1. Introduction

Relative permittivity of a material, $\epsilon_{r}$, is an important parameter for the design of antennas and microwave circuits because it strongly affects the dimensions of the circuit and the resonant frequency. When industry-standard dielectrics are used in the circuit design, their characteristics, including the relative permittivity, are known from the manufacturers’ data sheets. Nowadays, however, modern designs can also include some nonstandard materials, such as various fabrics (to be used in wearable applications, for example), whose permittivity is typically not included in the manufacturer’s specification because that parameter historically had no significance for the textile industry. Not knowing the permittivity introduces a significant uncertainty to the circuit designer, and it is necessary to determine it for a successful circuit design.

Various methods with very distinct features have evolved in that sense [1,2,3,4]—those that can characterize the materials under test in a wide frequency range, as opposed to those that do it in a narrow frequency band, practically at a single frequency; those that are invasive for the sample under test, to those that are noninvasive for the sample; those that are more suitable for low-frequency range, as opposed to those that are more suitable for high frequencies; those that are suitable for low-loss dielectric and thin samples vs. those that are suitable for liquid materials more than for solid materials; those with a more complicated measurement apparatus vs. those with a relatively simple one; those that measure the unknown material in a way that the test circuit is built using that unknown material as the substrate layer of the microstrip circuit, as opposed to those methods that use the known substrate layer and test the unknown material by laying it over, or inserting it in, the measurement fixture. There is no universal or best method to cling to, but the choice of a method really depends on one’s particular needs and case.

Commercial test kits are quite expensive, some even being compatible only with vector network analyzers (VNA) of the test kit vendor [5] or merely with some particular models of VNA and accompanying test fixtures (possibly of another vendor [6]), whereas some other vendors offer compatibility of their test kit with a broader range of VNAs [7], yet either of them offering only a conditional accuracy and prospect of getting accurate and trustworthy results due to the sensitivity of the measurement procedure to various tolerances. Facing these facts, a researcher with a limited budget may realize in a fairly short time that, for most practical purposes, it may be a sound option to build his own test circuit that will enable characterization of unknown dielectric materials.

In this work, it was desired to have a circuit that is simple to design and manufacture and, thus, cost-effective, while appropriate for characterization of thin, solid, and low-loss dielectrics. The frequency range of interest was below 6 GHz. More specifically, we aimed to use it at about 2.4 GHz. It is a single-frequency method, but that is not a penalizing factor due to the fact that permittivity changes little within a relatively wide frequency range and the test circuit can be designed and manufactured at a low cost for various single frequencies of interest. The particular circuit of our choosing was a *microstrip ring resonator* (MRR).

Amongst prior publications that were using an MRR structure, we found that in [8], a sample under test (SUT) was placed only over one gap, which is different from the majority of other works, and no permittivity-evaluation algorithm was enclosed. In [9], the algorithm was satisfactorily explained, but the SUT size and placement have not been unambiguously specified, nor were the SUTs measured therein clearly specified, except for one. In [10], a reflection-based approach was utilized (via $S_{11}$ parameter) instead of the transmission-based approach (via $S_{21}$ parameter), and the SUT was not characterized in the way we seek it here for characterizations of external SUTs. In [11], an MRR was built on the material that was meant to be characterized, rather than having an externally placed SUT, and the measurement procedure itself was not described. Likewise in [12], the relative permittivity of the substrate itself was characterized, which does not align with the objective in this work. In [13], a similar multi-layer structure for the characterization of permittivity was used, but only with simulations of the sample response and without the enclosed algorithm for the computation of the permittivity and measurements of real samples. In [14], only simulations were conducted, while the computation of $\epsilon_{r}$ was proposed in an indirect way, by setting a proportionality coefficient for the similar SUT, which is of limited applicability and reliability. Lastly, a comprehensive discussion on the permittivity evaluation was presented in [15], but the solution was based on a suspended ring structure, i.e., where the MRR is raised above the SUT, which does not enable a simple fabrication of the MRR.

The novelty and contribution of this paper are in the following:A novel measurement configuration based on the MRR circuit and the variational method is proposed and verified for determination of the relative permittivity of thin and solid dielectrics than has been defined before;It clearly presents the MRR design procedure with preparation of the proper SUT size and placement on the MRR surface, including the accompanying algorithm that does the ultimate evaluation of the SUT permittivity.

We declare this paper is a revised and extended version of our recent conference papers [16,17], now comprising new discussions and results, references, figures and tables, with a thorough discussion on the simplification to the previously defined measurement configuration.

The paper is organized as follows: Section 2 discusses the key aspects of the MRR-based approach. Section 3 presents the key expressions for the design of an MRR circuit. Section 4 describes the original MRR configuration and the algorithm. Section 5 discusses the proposed SUT dimensions to be used on the surface of the MRR. Section 6 is the focal part of the paper and presents proposed simplified measurement configuration with measurements of 12 benchmark materials, as well as discussions on accuracy and applicability. Lastly, Section 7 summarizes the most relevant contributions of the paper.

## 2. The Key Aspects of the MRR Method

Albeit the MRR circuit has been described in the literature, and the SUT measurement step, it is indeed a rare case to find a paper with a clear, complete, and unambiguous description of the SUT placement on the MRR surface and the SUT size (which matters!), as the accompanied algorithm to compute the $\epsilon_{r}$ from the obtained measurement results is often omitted with just the results presented. It is important to state that these three elements—the sample under test (SUT), the measurement configuration, and the algorithm to process the measured results—must be consistently and clearly defined for a successful implementation of this sensitive measurement. Two aspects are identified as a critical precondition for the success of this measurement principle:The MRR-based permittivity evaluation is based on measurement of the *unloaded* ($F_{0}$)- and *loaded* ($F_{1}$)-resonant frequency, where the former refers to an MRR without an SUT placed on its surface, while the latter refers to an MRR having an SUT placed on its surface. The ratio $F_{0}/F_{1}$ and the effective permittivity of the unloaded MRR ($\epsilon_{ef0}$) then provide the value of the effective permittivity of the loaded MRR ($\epsilon_{ef1}$). Yet, the effective permittivity is just the intermediate information being used, and the relative permittivity of the unknown sample is computed using an adequate algorithm. However, it turns out not to be such a straightforward task. In the open literature, it is actually a rare case to find a clearly described method of computation of the relative permittivity from the effective permittivity in this measurement configuration. So, while there is a wide consensus on how to design the MRR circuit, the evaluation of the relative permittivity of a material under test often remains hidden, while being presented only by tabulated results that a reader is supposed to trust (without being able to reproduce it from the enclosed description).The second crucial part for a successful evaluation of the unknown permittivity is getting a proper value of $F_{1}$, and that strongly depends on the *size* and *placement* of the SUT on the MRR surface. That aspect has been quite ambiguously covered in previous publications. Namely, the SUT size and its placement on the MRR surface greatly affect the measured value of $F_{1}$, which is then instrumental in the key equations to compute the $\epsilon_{r}$ of the SUT. Having SUTs of different sizes placed over the MRR surface, while not disclosing the accompanying equations to compute $\epsilon_{r}$, as has been the case in various earlier publications, does not really enable us reproduce the results and apply the approach with confidence. In particular, in ([8] Figure 2), the SUT was placed in the small area around one ring gap (the significance of the *gap* will be explained in more detail here in Section 3), with no expressions that served to evaluate the relative permittivity value based on measurements therein. In ([9] Figure 2), it seems that the SUT is placed over the entire surface of the substrate, yet it was not explicitly stated in the paper but remains open for a reader’s speculation. On the other hand, it is one of the rare papers, using MRR, where the computational model to evaluate $\epsilon_{r}$ was defined. In [13], measurements were neither conducted (even though the word “measurement” is declared in the paper title) nor the expressions for the computation of $\epsilon_{r}$ presented. In [14], the evaluation of the relative permittivity of several standard laminates was tested by full-wave simulations and establishing a characteristic *MRR factor* (see *D* in Equation (Equation 8), therein) that is pertinent to respective measurements. That approach can be helpful in situations with high-loss and/or thick SUTs (such as some food samples), where some other established algorithm does not produce good enough results, but it is essentially not a realiable method because we always first need a sample of a known permittivity value in order to determine that *MRR factor* for other samples, and the measured sample then generally must fall within similar values as the material sample used for the calibration. An interested reader can see from ([14] Tables 3–8) how the computed results for the same laminate depend from one calibration sample to another.

We experienced over the course of this work how $F_{1}$ strongly depends on the SUT size and placement. This issue, as well as the measurement and calculation procedure, has been inadequately covered—we emphasize this matter in Section 5.

## 3. Design of a Microstrip Ring Resonator

Although the design procedure of an MRR (Figure 1) is adequately described in the literature [13,15,18], we briefly include it for self-sufficiency of the paper.

### 3.1. Calculation of the Feed Line Width, Length, and the Ring Parameters

For the PCB substrate height *h*, we take that w/h>1 holds, which is typically the case, for which the line width *w* can be found [19] by iteratively solving the two coupled Equations (Equation 1) and (2)
(1)$$\begin{array}{cc} \epsilon_{eff} & =\frac{\epsilon_{r}+1}{2}+\frac{\epsilon_{r}-1}{2}\;{\left(1+12\frac{H}{w}\right)}^{-1/2} \end{array}$$
(2)$$\begin{array}{cc} Z_{0} & =\frac{120\;\pi }{\sqrt{\epsilon_{eff}}\;\left[\frac{w}{H}+1.393+\frac{2}{3}\ln\left(\frac{w}{H}+1.444\right)\right]} \end{array}$$

Alternatively, instead of utilizing the two equations given by (Equation 1) and (2), *w* can also be found by a single equation only, as defined in ([20] p. 145).

As for the ring, the essential idea is that its *mean* circumference (2πr) be equal to an integer multiple of the guided wavelength $\lambda_{g}$, that is
(3)$$2\pi r=n\lambda_{g}$$

Typically, the dominant mode of the operation is chosen, for which n=1. The guided wavelength $\lambda_{g}$ is computed as
(4)$$\lambda_{g}=\frac{c}{f_{r}\;\sqrt{\epsilon_{eff}}}$$
where *c* is the velocity of light, $f_{r}$ is the resonant frequency of the MRR structure, and $\epsilon_{eff}$ is the effective permittivity of the structure.

The feed line that is placed to the left and right of the ring is set to have its length $L_{f}$ equal to
(5)$$L_{f}=\lambda_{g}/4$$

Now, combining (Equation 3) and (Equation 4), the *mean* radius of the ring follows as
(6)$$r=\frac{c}{2\pi \;f_{r}\;\sqrt{\epsilon_{eff}}}$$
while the values of the *inner* and the *outer* ring radii are
(7)$$\begin{array}{c} r_{1}=r-\frac{w}{2} \end{array}$$
(8)$$\begin{array}{c} r_{2}=r+\frac{w}{2} \end{array}$$
where *w* is the microstrip line width (both for the feed line and the ring), as calculated by (Equation 1) and (2).

### 3.2. Calculating the Coupling Gap Width

In Figure 1, a small gap *g* between the feed line and the ring, on either side of the ring, can be observed. Together with the ring inductance, this capacitive gap creates a resonance at a certain frequency. As a rule of thumb for the optimal value of the gap width, some references recommend g≤1mm, and in ([15] p. 14), g∈[0.1w⋯w] is recommended. We determined its value by a CAD simulation (Figure 2) and making a compromise between the highest power transfer between the two ports and the value that will not be too small for the manufacturing of the circuit.

### 3.3. Manufactured MRR

Since the manufacturer’s PCB substrate was based on FR4 and we wanted to design our MRR at 2.45 GHz, it resulted in the design values listed in Table 1. The MRR shown in Figure 3 was manufactured at a professional facility [21]. Due to rounding the values of the line width, as well as the inner and the outer radii, to values that are more practical for manufacturing, the ultimate resonant frequency of the unloaded MRR, $F_{0}$, is slightly higher than the designed one, but that is irrelevant for the measurements because the result depends on the $F_{0}/F_{1}$ ratio (see (Equation 18) for details).

## 4. The Original MRR Configuration and the Algorithm

To evaluate relative permittivity of a SUT, a variational method-based algorithm was employed [9], as a seldom one that had a sufficiently described measurement and computation procedure, when it comes to the MRR method, but still without a clear reference to the size and placement of the SUT over the MRR surface, as we discussed in Section 2, nor having a clear reference to the SUTs that were tested.

### 4.1. The Original MRR-Based Measurement Structure

The variational method-based algorithm supports the MRR structure shown in Figure 4. It is structured as follows: on top of a ground plane, there is a substrate layer of permittivity $\epsilon_{1}$ and thickness *H*, as a typical microstrip-based configuration. On top of it is a conducting ring (of a negligible thickness *t*). In the next layer comes a sample under test (SUT) of an unknown permittivity $\epsilon_{2}$ and thickness *S*. The SUT is then covered by another dielectric of known permittivity $\epsilon_{3}$ and thickness *D*, which has its top surface covered by another ground plane.

Such a configuration [9] is adapted from ([22] Figure 1) in a way that the top and bottom dielectric layers of the configuration in [22] are swapped, which required the respective substitutions in the original algorithm such that $\epsilon_{1}↔\epsilon_{3}$ and H↔D.

### 4.2. The Variational Method-Based Algorithm

The algorithm explained to a greater detail can be found in [17], while here, we include only the key expression for easier reading of this text. To evaluate an SUT permittivity $\epsilon_{2}$, the *effective permittivity*$\epsilon_{f}$ of an MRR is to be computed by
(9)$$\epsilon_{f}=\frac{C(\epsilon_{1},\epsilon_{2},\epsilon_{3},H,S,D)}{C_{0}}$$
where *C* is the capacitance of an MRR with some specific non-air substrate, and $C_{0}$ is the capacitance of the MRR with air as substrate. The capacitance *C* [22,23] has to be computed a few times by
(10)$$\frac{1}{C}=\frac{1}{\pi \;Q^{2}\;\epsilon_{0}}\;\int_{0}^{\infty }{\left[\tilde{f}\left(\beta \right)\right]}^{2}\;\tilde{g}\left(\beta \right)\;\tilde{h}\left(\beta \right)\;d\beta$$
with the following substitutions
(11)$$\begin{array}{c} \frac{\tilde{f}\left(\beta \right)}{Q}=\frac{8}{5}\left\{\frac{\sin(\beta w/2)}{\beta w/2}\right\}+\frac{12}{5\;{\left(\beta w/2\right)}^{2}}\\ \cdot \left\{\cos\left(\beta w/2\right)-\frac{2\sin(\beta w/2)}{\beta w/2}+\frac{\sin^{2}\left(\beta w/4\right)}{{\left(\beta w/4\right)}^{2}}\right\} \end{array}$$
(12)$$\begin{array}{c} \tilde{g}\left(\beta \right)=\frac{\epsilon_{3}\;d+\epsilon_{2}\;s}{\left|\beta |\;\{\;\epsilon_{3}\;d\;\left[\;\epsilon_{1}\;h+\epsilon_{2}\;s\right]+\epsilon_{2}\;\left[\;\epsilon_{2}+\epsilon_{1}\;h\;s\right]\right\}} \end{array}$$
(13)$$\begin{array}{cc} d & =\coth(|\beta |\;D) \end{array}$$
(14)$$\begin{array}{cc} s & =\coth(|\beta |\;S) \end{array}$$
(15)$$\begin{array}{cc} h & =\coth(|\beta |\;H) \end{array}$$
(16)$$\tilde{h}\left(\beta \right)=\frac{1}{2}\left\{1+\frac{\sin H(|\beta |d-|\beta |t)}{\sin H(|\beta |d)}\right\}$$
where *t* is the line thickness, $\epsilon_{0}=8.854\times 10^{-12}\;\frac{F}{m}$ is the free-space permittivity, and *w* is the width of the ring and the feed line. Taking that t→0, $\tilde{h}\left(\beta \right)\to 1$ and (Equation 10) is computed by a numerical integration to successively obtain the capacitance of an air-filled line, $C_{0}$, capacitance of the unloaded MRR, $C_{1}$, and MRR structure capacitance *C* for various possible SUT permittivities $\epsilon_{2}$. That is obtained by
(17)$$\epsilon_{f0}=\frac{C_{1}}{C_{0}}\;$$
and
(18)$$\epsilon_{f1}=\epsilon_{f0}\;{\left(\frac{F_{0}}{F_{1}}\right)}^{2}$$
where $F_{0}$ and $F_{1}$ are the measured resonant frequency of the unloaded and loaded MRR, respectively. Then, by using (Equation 9) for $\epsilon_{f}\equiv \epsilon_{f1}$, $\epsilon_{2}$ is evaluated.

### 4.3. Initial Validation of the Algorithm

Initially, the algorithm was tested using various SUTs by a few approaches, as found in some earlier publications:Placing an SUT only over one ring gap (e.g., as in ([8] Figure 2));Placing an SUT over the whole ring and gaps (e.g., as in ([14] Figure 1b), varying the SUT sizes;Placing the SUTs over the entire area of the MRR, as it may be inferred from ([9] Figure 2).

The obtained results turned out to be inaccurate and unpredictable with respect to known SUT permittivities. Clearly, the results were quite sensitive to the size and placement of the SUT on the MRR surface, which required a more systematic analysis *to find the optimal size and placement of the SUTs* to achieve dependable measurement results. Apparently, the resonant frequencies $F_{0}$ and $F_{1}$ and the effective permittivity $\epsilon_{ef0}$ are the three factors that determine the ultimate result. Among them, $\epsilon_{ef0}$ is determined by analytical computation (without measurements) and $F_{0}$ has a repeatable value and a good agreement with the simulation. The *key factor* that affects the final result is $F_{1}$, which directly depends on the SUT size and placement on the MRR surface. More specifically, it was experienced that $F_{1}$ greatly depended on the sample width and length and whether we covered the entire surface of the resonator, or just the ring and the two gaps, or merely a fraction of the ring around one gap.

## 5. In Search of a Proper SUT Size

We therefore conducted full-wave analyses using some known laminate materials (whose data are available from data sheets) and evaluated the relative permittivity for every simulation, as was described in Section 4.2 [17]. In every simulation, the resulting resonant frequency $F_{1}$ was read off the $S_{21}$ curve. From the initial measurement of $F_{0}$ and the computed $\epsilon_{ef0}$, the effective permittivity of the loaded MRR was readily obtained by (Equation 9) from which the SUT permittivity, $\epsilon_{2}$, was computed. The simulated results were then verified by measurements using a VNA and the prepared SUTs.

### 5.1. Initial Examinations of the Proper SUT Size and Placement by Full-Wave Simulations

For the simulations, models of a TLX8 laminate [24] and a FR4 laminate were used, with the respective parameters values given in Table 2.

The simulations were performed using Altair Feko CAD solver [25], wherein we obtained $F_{0}=2503\;\mathrm{MHz}$. Initially, several variants of the SUT sizes were tested (Figure 5) to be concluded that the SUT should be of the approximate size to cover the ring including both gaps. It was also observed that more accurate results occurred when the SUT width was $SUT_{W}\geq 2r_{2}$ and more detailed analyses were needed to establish a more dependable condition on the SUT size.

### 5.2. Detailed SUT Size Examinations by Full-Wave Simulations

In the following simulations, either the SUT length ($SUT_{L}$) or width ($SUT_{W}$) were varied one at a time, with the specific settings that are presented in Table 3, Table 4 and Table 5.

In Table 3, where the sample length, $SUT_{L}$, was varied from $SUT_{L}=L_{g}$ down to $SUT_{L}=2\left(r_{2}+g+1\right)$, while keeping the sample width constant at the full width ($SUT_{W}=W_{g}$), we see that the closest result to the expected permittivity value of TLX8 ($\epsilon_{2}=2.55$) is in the last column (i.e., where the length of the SUT is just slightly wider than the ring width plus the gap width) and equals $\epsilon_{2}=2.57$.

The next series of simulations, shown by Table 4, was characterized by also varying the sample length, analogously to the Table 3, but now with a *reduced sample width* ($SUT_{W}=2\;r_{2}$).

We can observe similar, yet slightly worse, results in this case, with the relatively most accurate result in the last column, just as in Table 3. It is noticeable that the results are less accurate when the sample length is larger (e.g., as in columns 2, 3, and 4). It can be interpreted by the notion that having a sample covering a longer fraction of the feed line alters the effective permittivity, which consequently affects the computed relative permittivity of the unknown sample $\epsilon_{2}$.

In the next analysis, summarized in Table 5, the sample length was kept constant at $SUT_{L}=2\;\left(r_{2}+g+1\right)$, while varying the sample width, $SUT_{W}$.

What can be observed from the respective tabulated results is that the most accurate result occurs for the full width of the SUT, (i.e., $SUT_{W}=W_{g}$), which is $\epsilon_{2}=2.57$ (in column 2), but the result remains quite steady down to $SUT_{W}=2\;r_{2}$, with some fluctuation around $SUT_{W}=0.6\;W_{g}$, and becomes inaccurate but for $SUT_{W}<2\;r_{2}$ (see the rightmost column where $SUT_{W}=r_{1}$), which means that the outer edge of the ring must be included in the SUT size. The best result was $\epsilon_{2}=2.57$, which is only 0.78% or relative error with respect to the nominal value for this TLX8 laminate, which is $\epsilon_{n}=2.55$, and that is an excellent result for such a simple measurement apparatus. If we question why the results are less sensitive to varying the SUT width as opposed to varying the SUT length, the likely reason lies in the fact that varying the SUT width does not cross the feed line and hence does not affect the structure’s effective permittivity and line impedance as much.

Ultimately, the following SUT dimensions were found to be proper: (19)$$\begin{array}{cc} SUT_{L} & =2\;(r_{2}+g+1) \end{array}$$(20)$$\begin{array}{cc} SUT_{W} & =W_{g} \end{array}$$

## 6. The Proposed Simplified MRR Configuration for Measurement of SUT Permittivity

In our earlier work ([17] Table 1), it was experienced that the results obtained by the original MRR configuration (as in Figure 4) were not desirably close to the reference values of SUT permittivity ($\epsilon_{2}$). In addition, working with a PTFE block, as the cover layer for the SUT, is tricky because of its very slippery surface. (We also tried to use a different dielectric as the cover layer, but it brought no benefit in terms of the results.)

In light of that experience, the idea was contemplated if a simplified MRR structure could support the SUT measurements as well. The essence of that simplification was to remove the layers above the SUT, as illustrated in Figure 6. Removal of the cover layer is then adequately reflected in the variational method-based algorithm by having $\epsilon_{3}=1$ and D→∞.

Without the cover layer, it is now easier to align the SUT with the MRR board, but care has to be taken to make a firm contact between the SUT and the MRR surface. The sensitivity of the measured results, with respect to the pressure applied on the SUT surface, was also initially tested in [17] by three ways:Placing an SUT over the MRR surface without applying any pressure;Pressing the SUT using a pair of plastic clamps (Figure 7a);Pressing the SUT using just fingers (Figure 7b).

The accuracy of the results showed improvement (see ([17] Table 2)), wherein the best results were actually achieved when the SUT surface was pressed with fingers (Figure 7b). For TLX8, the relative error was only −3.9%, while for the jeans sample, the exact permittivity value was achieved. (In the open literature, permittivity value of jeans varies from 1.6 to 1.8 [26,27,28,29], with the prevailing value of 1.7.) However, the above findings called for additional verification.

### 6.1. Detailed Evaluation of the Impact of Finger-Pressing on the SUT Surface

We understand that applying direct finger-pressing on the surface of an SUT may raise a doubt, for the reason that the SUT is no longer covered by an even layer that is bounded by the upper ground plane, as in the *original* configuration, and that touching the SUT surface with fingers introduces some uncertainty to the measurement. Additional verification is also important to achieve confidence in the repeatability of the measurement results and that is especially important because we are dealing with an open structure without a strict enclosure for the SUT placement, plus we propose it to be possible to remove the cover layer. As the fingers can be placed across the MRR surface in various ways, it is important to check whether some finger-pressure configurations are better than the others with respect to the accuracy of the results.

To provide more insights into that question, we performed more detailed measurements using four configurations of fingers placed on the SUT surface, as illustrated by Figure 8, to determine the one that is realistic and practical enough to perform the measurements and provide the most accurate and consistent results.

The measurements were performed on multiple samples (Figure 9) of various SUT thicknesses (*S*), ranging from 0.1 mm up to over 7 mm, and permittivity values ($\epsilon_{n}$), ranging from 1.7 to over 6. The measurement results are presented in Table 6 in terms of the measured $F_{1}$ and the computed relative permittivity value of each SUT ($\epsilon_{x}$).

During these measurements, it became clear that 3-finger and 5-finger configurations did not secure accurate enough results and. due to that, only several SUTs were measured by those two configurations (i.e., the dashes in the table mean that the measurement in that case was not performed). We interpret such an outcome by the presence of a finger within the radius of the ring, possibly near the strongest field lines. As for the comparison between the 2-finger and 4-finger configurations, the results are almost equal for the thicker SUTs (e.g., the last four rows of Table 6), while their proximity to the nominal value of the SUT permittivity ($\epsilon_{n}$ in column 3) varies for thinner SUTs. Still, both results are satisfactorily close to the reference value of the respective permittivities.

We now extract the tabular values for the 2-finger and 4-finger configurations and express the relative errors with respect to the nominal permittivity value for the given SUTs. The results are shown in Table 7.

It can be observed through three groups of SUT thicknesses that are separated by horizontal lines. The first group contains very thin samples like 0.1-mm thin paper. At first, the accuracy was inadequate. However, when two layers of the sample were stacked together to double the thickness, the 4-finger configuration achieved the accurate value of $\epsilon_{r}$. In this case, four fingers helped stick the sample more evenly to the MRR surface and reduce possible air bubbles as a result of an easily bendable surface of the paper.

The second group of SUTs comprises five industry-standard laminates (TLX’s, RF60A’s, and FR4) and two non-standard SUTs (jeans and glass), all with moderate sample thicknesses from 0.8 mm to 1.93 mm and permittivities ranging from 1.7 to 7. For this group of samples, we see that both 2-finger and 4-finger setup achieved close results and all with relative errors below 10%, which is a great achievement for one such simple measurement setup. We note that the declared nominal values $\epsilon_{n}$ were copied from the official datasheets of the respective laminate manufacturers or from the sources available in the open literature. Each item in Table 7 was added a reference to the source of information. For those SUTs that are known to not have a unique value of $\epsilon_{n}$, such as FR4 or glass, but can come within a range of values, we did not calculate the relative error, but presented only the computed permittivity value $\epsilon_{x}$ for each case. It can be observed that our results fall within the nominal range of values for both of these SUTs. It may also be a good practice to measure a given SUT by both the 2-finger- and 4-finger- configurations and then average the result, which will secure a modest distance from the exact values of the SUT permittivity, whichever of the two measurements happened to be more accurate in the particular case.

The last group of SUTs are thicker samples, such as three available variants of plexiglass material, with three different thicknesses, and a sample of PTFE. While PTFE has quite a unanimous permittivity value about 2 [32], plexiglass SUTs are found to have $\epsilon_{n}$ within a range from 2.6 to 3.5 [31]. For all the three types of plexiglass, the measured results were close to the lower end of the nominal $\epsilon_{n}$, while for the PTFE, the evaluated $\epsilon_{x}$ was 1.67, which is 17% below the nominal value. In case of the last two SUTs, i.e., the plexiglass and PTFE, the discrepancy is somewhat higher. The percentage value in this case looks more concerning, though, than we feel looking at the absolute difference between $\epsilon_{x}$ and $\epsilon_{n}$. Nevertheless, it is generally understood that the resonant method is primarily good for *thin* samples. For thicker samples, sensitivity of the MRR method decays, due to which the frequency shift of $F_{1}$ is not as accurate as in the case of thinner SUTs.

### 6.2. Why a Direct Contact of Fingers with the SUT Does Not Spoil the Measurement

The concern with this simplified measurement, by a direct finger touch with the SUT surface, a direct touch with the SUT surface, may be that it prohibitively affects the measurement. Although this concern is justified, our unbiased measurements and computations of SUTs permittivity, which were conducted by both approaches (i.e., with- and without- the cover layer) indicated no detrimental effect to the ultimate result when the “simplifed” configuration from Figure 6 was used instead of the “original” MRR configuration in Figure 4. In Table 8, we directly compare the results obtained by using the *simplified* configuration vs. *original* configuration, as illustrated by Figure 10.

From Table 8, it is evident that measurements performed by the simplified MRR configuration actually achieved better results with all the measured SUTs. In the most relevant group of *moderately thin* SUTs, the simplified method outperformed the original even by the double digits in percentages of the relative error. Now, how can we explain such an outcome? When we look at the surface current over the MRR surface, as shown in Figure 11, we see that the spots where the fingers in the “2-finger” and “4-finger” configurations are being placed have surface current levels many dBs below the strongest surface current. Due to that, the finger placement on the SUT surface has an insignificant effect on the measured result and is acceptable for the practical purposes.

Alternatively, if we imagine the use of an infinitely thick cover layer, the ground plane on it will not have an impact as it has with a finite thickness of the layer. We consider that a model of an infinitely thick cover layer of air without the ground plane on it then has an equivalent effect that is what was entered in the algorithm by setting D→∞ and $\epsilon_{3}=1$.

Another aspect is measurement accuracy and reading precision in the VNA software, as for the ultimate computation of $\epsilon_{2}$, every MHz counts when computing expression (Equation 18). To illustrate it, Figure 12 shows how the results of $\epsilon_{f1}$ and $\epsilon_{2}$ vary with respect to a possible result of $F_{1}$. For example, let us take $F_{1}=2346\;\mathrm{MHz}$. In Figure 12, we show the values of $\epsilon_{f1}$ and $\epsilon_{2}$ if $F_{1}$ were ±5MHz around the reported value of 2346 MHz. Thus, $F_{1}$ values span from $F_{1}-5\;\mathrm{MHz}=2341\;\mathrm{MHz}$, for which $\epsilon_{2}=2.66$, to $F_{1}+5\;\mathrm{MHz}=2351\;\mathrm{MHz}$, where $\epsilon_{2}=2.53$.

The change in the results of $\epsilon_{f1}$ and $\epsilon_{2}$ for the lowermost frequency point (2341 MHz) with respect to the central point (2346 MHz) is the following: the relative discrepancy in $F_{1}$ is −0.21%, for which the discrepancy in the value of $\epsilon_{f1}$ is 0.43%, but the discrepancy in the value of $\epsilon_{2}$ is 2.7%. It speaks something about the sensitivity of the result to the measured values because, for a small shift in the measured frequency value $F_{1}$ of merely 0.21%, the change in the resulting value of $\epsilon_{2}$ was almost 3%. Knowing that, it is fair to conclude that the results achieved by this simple measurement apparatus are quite satisfactory for ocassional measurements of unknown dielectric sheets that satisfy the characteristics of this measurement method.

## 7. Conclusions

The focal part of this paper was the proposed simplified configuration for measurement of thin dielectric materials in conjunction with a variation method-based algorithm. We presented the modification in the measurement setup, showed the essence of the algorithm that post-processes the measured values, and we verified it on 12 dielectric samples of different thicknesses and permittivities, with accuracies below 10% of relative error for all thin sheets and moderately higher errors (below 20%) for thick sheets that are generally not considered suitable for measurement with this method.

For completeness of the presentation, the MRR design equations and the key expressions of the variational algorithm were included to a degree that provides a self-sufficiency of this text. Additionally, a discussion on proper SUT size and placement on the MRR surface was included to emphasize the importance of it for overall success of the measurement and evaluation of an unknown permittivity.

The novelty of this work is in the proposition of a measurement configuration that is simpler than the one previously defined, while the contribution is that it includes the circuit design, permittivity computation expressions, and a recommendation for the proper sample size preparation, in addition to the proposed measurement approach.

## Figures and Tables

**Figure 1 sensors-22-00928-f001:**
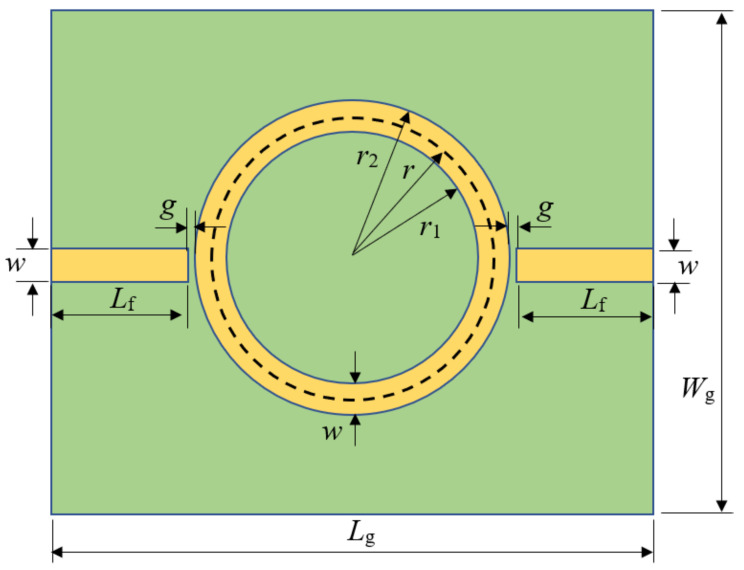
The layout of the microstrip ring resonator (MRR) circuit for characterization of an unknown material permittivity.

**Figure 2 sensors-22-00928-f002:**
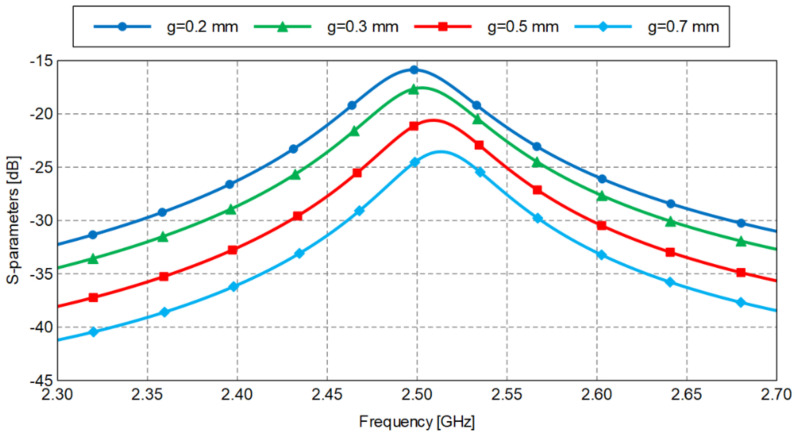
Optimization of the gap value by a full-wave simulation.

**Figure 3 sensors-22-00928-f003:**
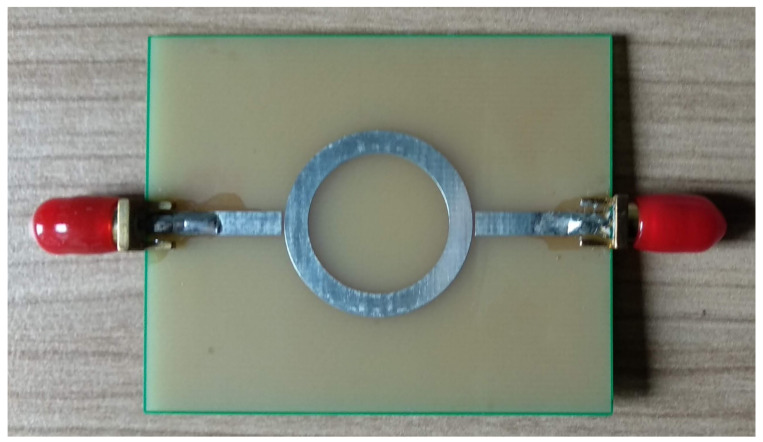
The MRR manufactured at JLCPCB factory.

**Figure 4 sensors-22-00928-f004:**
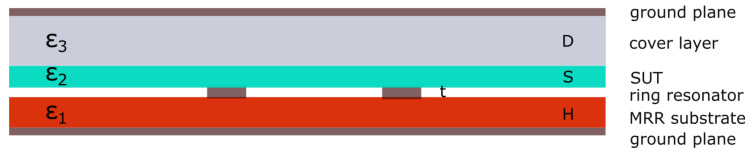
The original MRR configuration as used in [9].

**Figure 5 sensors-22-00928-f005:**
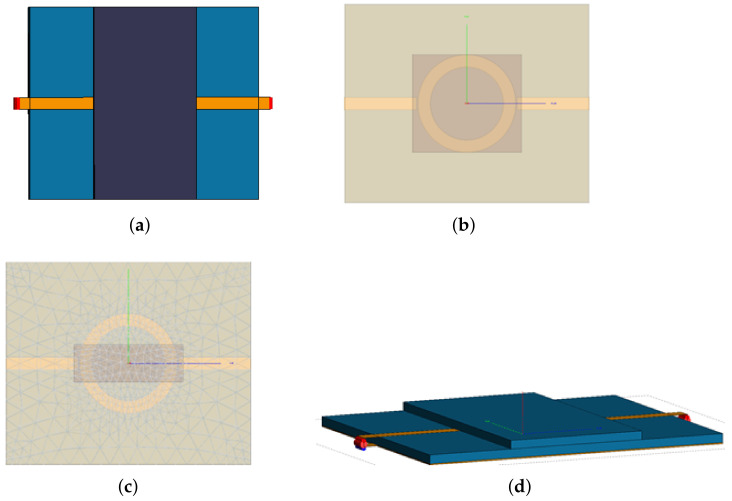
Various SUT variants of the early examinations. (**a**) TLX8 SUT: full width. (**b**) TLX8 SUT: medium width. (**c**) TLX8 SUT: short width. (**d**) FR4 SUT: full width.

**Figure 6 sensors-22-00928-f006:**
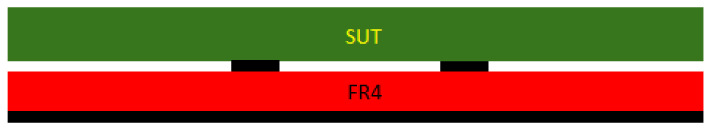
The cross-section of a simplified MRR configuration—without the “cover” layer on top of an SUT.

**Figure 7 sensors-22-00928-f007:**
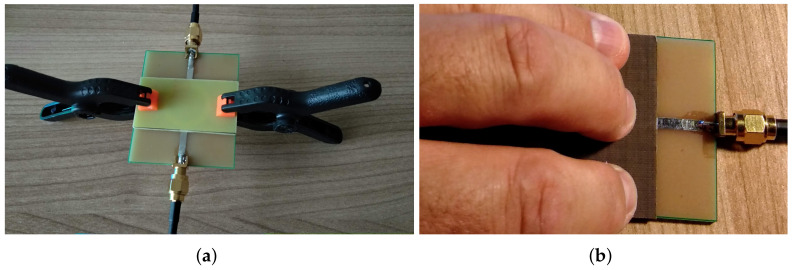
Measurements on FR4 with clamps and TLX8 with fingers. (**a**) measurement of FR4. (**b**) measurement of TLX8.

**Figure 8 sensors-22-00928-f008:**
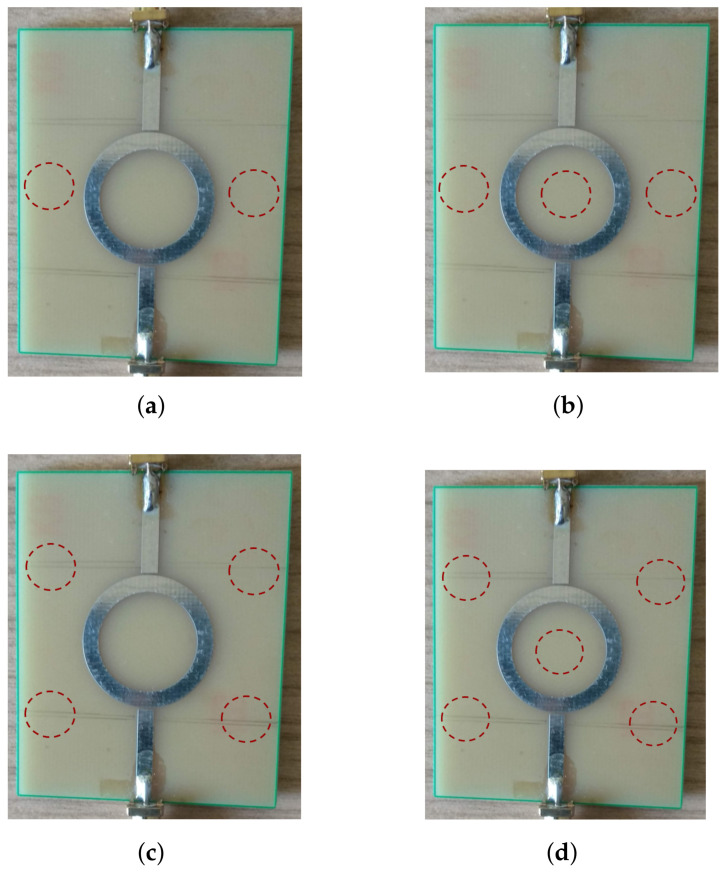
The four finger-press configurations with respective finger spots for the surface of an SUT. (**a**) 2-finger configuration. (**b**) 3-finger configuration. (**c**) 4-finger configuration. (**d**) 5-finger configuration.

**Figure 9 sensors-22-00928-f009:**
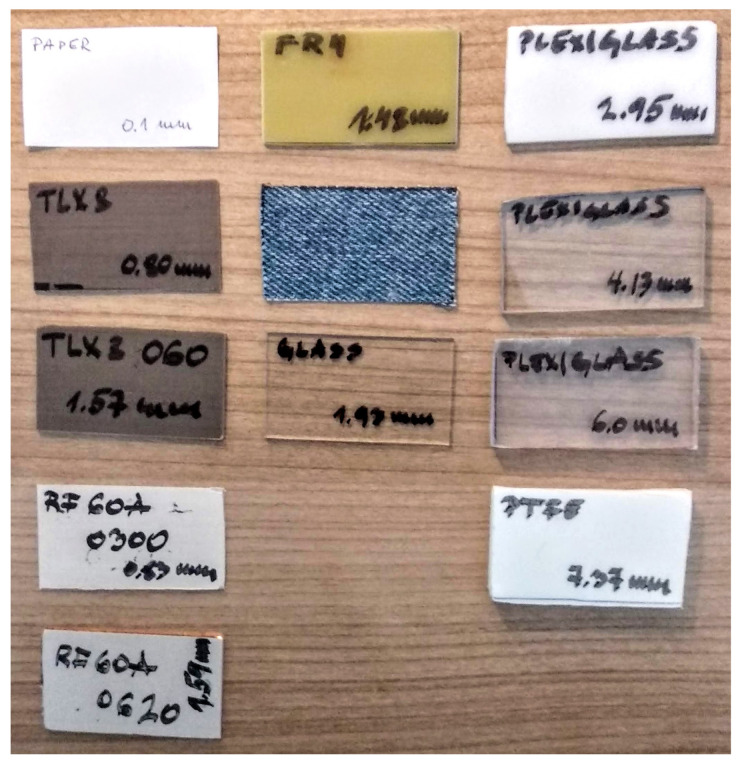
Dielectric materials that were tested.

**Figure 10 sensors-22-00928-f010:**
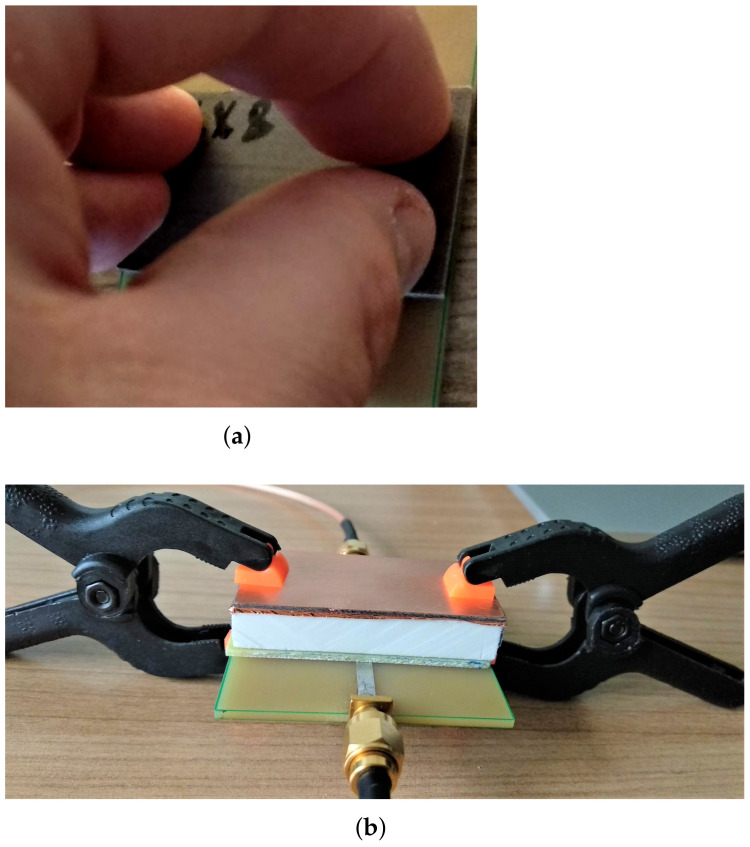
Comparative measurements of SUTs by “simplified” MRR configuration vs. the “original” MRR configuration with a PTFE cover layer and a ground plane on top. (**a**) “4-finger” simplified configuration. (**b**) original configuration.

**Figure 11 sensors-22-00928-f011:**
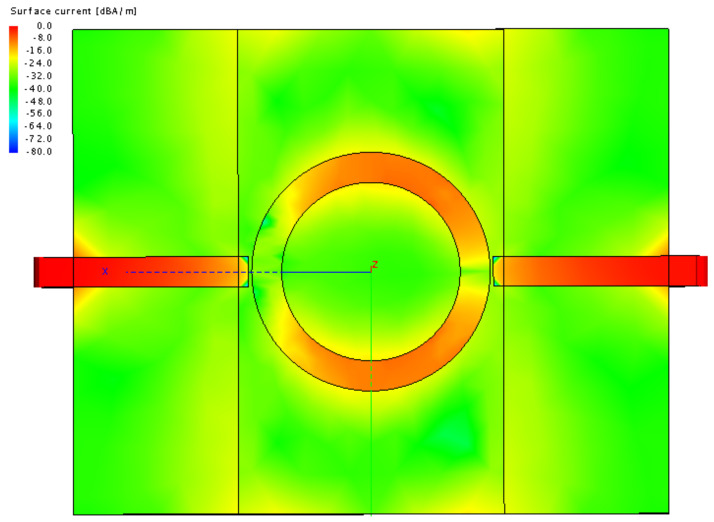
The simulated surface current.

**Figure 12 sensors-22-00928-f012:**
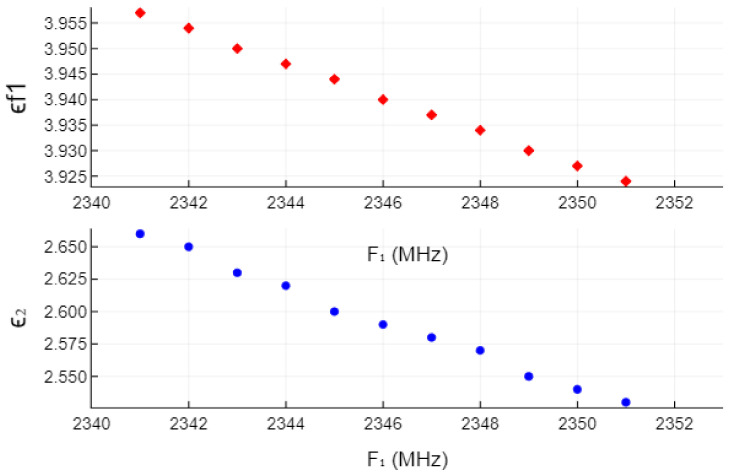
An example of change of the evaluated value of $\epsilon_{f1}$ and $\epsilon_{2}$, with respect to $F_{1}$, when $F_{1}$ varies ±5MHz around the central value at 2346 MHz for an SUT of 4.13 mm thick plexiglass.

**Table 1 sensors-22-00928-t001:** Design parameters of our MRR.

Parameter	Value
Design resonant frequency, $F_{0}$	2.45 GHz
MRR substrate height (FR4), *H*	1.55 mm
MRR substrate permittivity (FR4), $\epsilon_{1}$	4.6
Design parameters adjusted for manufacturing	
Feed line width, *w*	2.9 mm
Inner ring radius, $r_{1}$	8.6 mm
Outer ring radius, $r_{2}$	11.5 mm
Feed line length, $L_{f}$	16.5 mm
Mean ring radius, $r=(r_{1}+r_{2})/2$	10.05 mm
Coupling gap, *g*	0.3 mm
PCB length, $L_{g}=2\;\left(L_{f}+r_{2}+g\right)$	56.6 mm
PCB width, $W_{g}$	46.0 mm

**Table 2 sensors-22-00928-t002:** Sample under test (SUT) values used in simulations.

SUT Material	SUT Thickness, S	Nominal SUT Permittivity, $\epsilon_{2}$
TLX8	0.82 mm	2.55
FR4	1.53 mm	4.3 ...4.6 *

* The actual value of FR4 depends on the manufacturer.

**Table 3 sensors-22-00928-t003:** Analyzing resonant frequencies for full sample width (${\mathrm{SUT}}_{W}=W_{g}$) and variable sample length, $SUT_{L}$.

Sample Length *, ${\mathrm{SUT}}_{L}$	$L_{g}$	$0.65\;L_{g}$	$0.60\;L_{g}$	$0.55\;L_{g}$	$2\;(r_{2}+g+1)$
$F_{1}$ (MHz)	2405	2405	2401	2407	2409
$\epsilon_{ef1}$	3.743	3.743	3.755	3.736	3.73
$\epsilon_{2}$	2.65	2.65	2.72	2.6	**2.57**

* with: *ε*_n_ = 2.55 mm and *S* = 0.82 mm for a TLX8 SUT, and *F*_0_ = 2503 MHz.

**Table 4 sensors-22-00928-t004:** Analyzing resonant frequencies for reduced sample width (${\mathrm{SUT}}_{W}=2\;r_{2}$) and variable sample length, $SUT_{L}$.

Sample Length *, ${\mathrm{SUT}}_{L}$	$L_{g}$	$0.65\;L_{g}$	$0.60\;L_{g}$	$0.55\;L_{g}$	$2\;(r_{2}+g+1)$
$F_{1}$ (MHz)	2402	2404	2397	2403	2408
$\epsilon_{ef1}$	3.75	3.746	3.77	3.749	3.733
$\epsilon_{2}$	2.72	2.68	2.83	2.69	**2.6**

* with: *ε*_n_ = 2.55 mm and *S* = 0.82 mm for a TLX8 SUT, and *F*_0_ = 2503 MHz.

**Table 5 sensors-22-00928-t005:** Analyzing resonant frequencies for constant sample length ($SUT_{L}=2\;\left(r_{2}+g+1\right)$) and variable sample width, $SUT_{W}$.

Sample Width *, ${\mathrm{SUT}}_{W}$	$W_{g}$	$0.8\;W_{g}$	$0.7\;W_{g}$	$0.65\;W_{g}$	$0.6\;W_{g}$	$0.55\;W_{g}$	$2\;r_{2}$	$r_{1}$
$F_{1}$ (MHz)	2409	2409	2409	2409	2411	2408	2408	2451
$\epsilon_{ef1}$	3.729	3.279	3.279	3.279	3.724	3.733	3.733	3.604
$\epsilon_{2}$	**2.57**	2.57	2.57	2.57	2.53	2.6	2.6	1.8

* with: *ε*_n_ = 2.55 mm and *S* = 0.82 mm for a TLX8 SUT, and *F*_0_ = 2503 MHz.

**Table 6 sensors-22-00928-t006:** Measurements of the impact of the four finger configurations.

SUT	S, mm	$\epsilon_{n}$	2 Fingers		3 Fingers		4 Fingers		5 Fingers	
			$F_{1}$, MHz	$\epsilon_{x}$	$F_{1}$, MHz	$\epsilon_{x}$	$F_{1}$, MHz	$\epsilon_{x}$	$F_{1}$, MHz	$\epsilon_{x}$
paper: 1 layer	0.1	3	2469	4.27	2429	9.17	2490	2.17	2463	4.92
paper 2 layers	0.2	3	2449	3.75	2366	9.84	2462	2.99	2334	n/a
TLX8	0.8	2.55	2407	2.7	2392	3.01	2405	2.74	2361	3.74
TLX8–060	1.57	2.55	2375	2.65	-	-	2380	2.57	-	-
RF60A–0300	0.83	6.15	2277	5.94	-	-	2268	6.2	-	-
RF60A–0620	1.59	6.15	2210	5.73	-	-	2205	5.84	-	-
jeans	0.9	1.7	2456	1.73	2435	2.07	2447	1.86	2436	2.05
FR4	1.48	4.3–4.6	2287	4.23	-	-	2276	4.44	-	-
glass	1.93	4–7	2174	6.09	-	-	2166	6.28	-	-
plexiglass 1	2.95	3.45	2371	2.4	-	-	2375	2.34	-	-
plexiglass 2	4.13	3.45	2350	2.55	-	-	2349	2.57	-	-
plexiglass 3	6	3.45	2351	2.47	-	-	2349	2.49	-	-
PTFE	7.37	2	2428	1.67	-	-	2428	1.67	-	-

With *F*_0_ = 2560 MHz, *D* ≫, *ε* = 1.

**Table 7 sensors-22-00928-t007:** The relative error comparison of the two best finger configurations.

SUT	S, mm	$\epsilon_{n}$	2 Fingers			4 Fingers		
			$F_{1}$, MHz	$\epsilon_{x}$	Δϵ	$F_{1}$, MHz	$\epsilon_{x}$	Δϵ
paper: 1 layer	0.1	3 [30]	2469	4.27	42%	2490	2.17	−28%
paper: 2 layers	0.2	3 [30]	2449	3.75	25%	2462	2.99	0%
TLX8	0.8	2.55 [24]	2407	2.7	6%	2405	2.74	7%
TLX8–060	1.57	2.55 [24]	2375	2.65	4%	2380	2.57	1%
RF60A–0300	0.83	6.15 [24]	2277	5.94	−3%	2268	6.2	1%
RF60A–0620	1.59	6.15 [24]	2210	5.73	−7%	2205	5.84	−5%
jeans	0.9	1.7 [26,27,28,29]	2456	1.73	2%	2447	1.86	9%
FR4	1.48	4.3–4.6	2287	4.23	-	2276	4.44	-
glass	1.93	4–7 [30]	2174	6.09	-	2166	6.28	-
plexiglass 1	2.95	2.6–3.5 [31]	2371	2.4	-	2375	2.34	-
plexiglass 2	4.13	2.6–3.5 [31]	2350	2.55	-	2349	2.57	-
plexiglass 3	6	2.6–3.5 [31]	2351	2.47	-	2349	2.49	-
PTFE	7.37	2 [32]	2428	1.67	−17%	2428	1.67	−17%

With *F*_0_ = 2506 MHz.

**Table 8 sensors-22-00928-t008:** Result comparison: a “simplified” vs. the “original” configuration.

SUT	S, mm	$\epsilon_{n}$	with PTFE Cover and Clamps			Simplified: 4 Fingers		
			$F_{1}$, MHz	$\epsilon_{x}$	Δϵ	$F_{1}$, MHz	$\epsilon_{x}$	Δϵ
paper: 1 layer	0.1	3	2411	NaN	-	2490	2.17	−28%
paper: 2 layers	0.2	3	-	-	-	2462	2.99	0%
TLX8	0.8	2.55	2389	1.84	−28%	2405	2.74	7%
TLX8–060	1.57	2.55	2369	2.15	−16%	2380	2.57	1%
RF60A–0300	0.83	6.15	2233	5.23	−15%	2268	6.2	1%
RF60A–0620	1.59	6.15	2195	5.04	−18%	2205	5.84	−5%
jeans	0.9	1.7	2421	1.42	−16%	2447	1.86	9%
FR4	1.48	4.3–4.6	2283	3.43	-	2276	4.44	-
glass	1.93	4–7	2189	4.92	-	2166	6.28	-
plexiglass 1	2.95	2.6–3.5	2371	2.13	-	2375	2.34	-
plexiglass 2	4.13	2.6–3.5	2358	2.28	-	2349	2.57	-
plexiglass 3	6	2.6–3.5	2361	2.26	-	2349	2.49	-
PTFE	7.37	2	2427	1.63	−19%	2428	1.67	−17%

With *F*_0_ = 2507 MHz.

## Data Availability

Not applicable.

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
