# Peer review of "A Simplified Measurement Configuration for Evaluation of Relative Permittivity Using a Microstrip Ring Resonator with a Variational Method-Based Algorithm"

_sensors, 2022, doi:10.3390/s22030928_

Round 1

Reviewer 1 Report

In this paper a Simplified Measurement Configuration for Evaluation of Relative Permittivity using a Microstrip Ring Resonator, is proposed, which present valuable work but, novelty of the proposed work is restricted and lot of parts of this work previously introduced by authors. The manuscript can be published after some minor modifications, which are listed as follows:

1-The title should be modified, the word “Optimal” should be revised.

2- The novelty and main contribution of the proposed work should be clearly emphasized.

3-  The abstract and conclusion are reported qualitatively, if possible to more clearness expressed them quantitatively with numbers and improved parameters in our work.

4- many parts of the manuscript are copied from [R1] and [R2], which are restricted the novelty of the proposed work. 

[R1] Miroslav Joler, Alex N. J. Raj, Juraj Bartolic. "Finding an Optimal Sample Size and Placement for Measurement of Relative Permittivity using a Microstrip Ring Resonator", 2021 International Conference on Software, Telecommunications and Computer Networks (SoftCOM), 2021

[R2] Miroslav Joler, Alex N. J. Raj. "Relaxing the Variational Method-based Measurement Configuration for the Evaluation of Permittivity using a Microstrip Ring Resonator", 2021 International Conference on Software, Telecommunications and Computer Networks (SoftCOM), 2021

Author Response

Dear Reviewer,

We thoroughly replied to the questions from you Review. The responses are contained in the PDF file we uploaded with the revised version.

Thank you for your time and valuable feedback!

Sincerely,

Miroslav Joler

Reviewer 2 Report

This paper presents a simplified MRR Configuration for Measurement of SUT Permittivity. I have several questions:

  • Please provide the compare of the presented "simplified MRR config" and to "traditional MRR config with copper ground".
  • As shown in Fig.8, the one without copper ground, how authors to eliminate the affect by tester clip or tester's hands? Without ground layer, this would add testing uncertainty and errors.
  • How to make sure the test sample has a close contact with the resonator?

Author Response

Dear Reviewer,

We thoroughly replied to all the questions from you review. The responses are contained in a PDF file we upload here.

We thank you for your time and valuable comments!

Sincerely,

Miroslav Joler

Round 2

Reviewer 1 Report

The authors have addressed all concerns. This manuscript can be accepted.

Reviewer 2 Report

All my previous questions have been answered!